# Isoprene Oxidation by the Gram-Negative Model bacterium *Variovorax* sp. WS11

**DOI:** 10.3390/microorganisms8030349

**Published:** 2020-02-29

**Authors:** Robin A. Dawson, Nasmille L. Larke-Mejía, Andrew T. Crombie, Muhammad Farhan Ul Haque, J. Colin Murrell

**Affiliations:** 1School of Environmental Sciences, Norwich Research Park, University of East Anglia, Norwich NR4 7TJ, UK; R.Dawson@uea.ac.uk (R.A.D.); n.mejia@uea.ac.uk (N.L.L.-M.); 2School of Biological Sciences, Norwich Research Park, University of East Anglia, Norwich NR4 7TJ, UK; A.Crombie@uea.ac.uk; 3School of Biological Sciences, University of the Punjab, Quaid-i-Azam Campus, Lahore 54000, Pakistan; mfarhan.sbs@pu.edu.pk

**Keywords:** isoprene, microbiology, monooxygenase, mutagenesis, isoprene-fed fermentor

## Abstract

Plant-produced isoprene (2-methyl-1,3-butadiene) represents a significant portion of global volatile organic compound production, equaled only by methane. A metabolic pathway for the degradation of isoprene was first described for the Gram-positive bacterium *Rhodococcus* sp. AD45, and an alternative model organism has yet to be characterised. Here, we report the characterisation of a novel Gram-negative isoprene-degrading bacterium, *Variovorax* sp. WS11. Isoprene metabolism in this bacterium involves a plasmid-encoded *iso* metabolic gene cluster which differs from that found in *Rhodococcus* sp. AD45 in terms of organisation and regulation. Expression of *iso* metabolic genes is significantly upregulated by both isoprene and epoxyisoprene. The enzyme responsible for the initial oxidation of isoprene, isoprene monooxygenase, oxidises a wide range of alkene substrates in a manner which is strongly influenced by the presence of alkyl side-chains and differs from other well-characterised soluble diiron monooxygenases according to its response to alkyne inhibitors. This study presents *Variovorax* sp. WS11 as both a comparative and contrasting model organism for the study of isoprene metabolism in bacteria, aiding our understanding of the conservation of this biochemical pathway across diverse ecological niches.

## 1. Introduction

Many volatile organic compounds (VOCs) influence atmospheric chemistry, typically as a result of their breakdown via physical and chemical processes. Isoprene accounts for one-third of biogenic VOCs emitted to the atmosphere, with production by terrestrial plants accounting for approximately 500 Tg C yr^−1^ [1]. This is comparable to methane in terms of global VOC production [1,2]. The atmospheric half-life of isoprene is short due to the reactivity of its two alkene groups [2]. The presence of the C=C double bonds makes isoprene prone to nucleophilic attack by hydroxyl radicals, and thus isoprene is an indirect contributor to global warming due to competition for hydroxyl radicals responsible for the removal of atmospheric methane [2]. The products of isoprene oxidation form secondary organic aerosols that can act as cloud condensation nuclei, causing the shielding effects which contribute to global cooling [3,4]. Isoprene also influences levels of ozone. NO_x_ levels associated with urban environments react with isoprene oxidation products, generating various compounds including O_3_ [2]. Conversely, ozone is depleted from low-NO_x_ areas due to the direct reaction of isoprene with O_3_ [5].

Isoprene synthase (IspS) is the key enzyme in biogenic isoprene production [6,7]. *ispS* is sporadically distributed in terrestrial plants, with varied roles being suggested for isoprene production including protection against heat and oxidative stress [8,9,10,11]. Despite the debate surrounding the biological role of isoprene, the significance of isoprene production in plants is clear when considering the metabolic costs. A single molecule of isoprene requires the input of 6 carbon atoms, 20 ATP and 14 NADPH [8] and thus, given that the production of isoprene typically uses 2% of fixed carbon [8], represents a significant investment by the tree. The roles of isoprene in abiotic stress responses in plants have been discussed by Vickers et al. [9,10]. Recent studies have demonstrated that isoprene influences gene expression in plants, as reviewed by Lantz et al. [12]. For example, *Arabidopsis thaliana*, a non-isoprene emitter, engineered to express *ispS* from *Eucalyptus globulus*, demonstrated regulation of isoprene emission and photosynthesis pathways which were comparable to the native emitter [11]. The protective role of isoprene against heat and light stress was also demonstrated when *Arabidopsis thaliana* was fumigated with isoprene [13]. The overlap of gene expression in the fumigation and transgenic expression models indicated a role of isoprene emission in the regulation of plant genes. Marine algae are another biogenic source of isoprene, with estimates ranging from 0.1 to 12 Tg C yr^−1^ compared to approximately 500 Tg C yr^−1^ for terrestrial plants [14,15,16]. Marine isoprene production can support growth of a variety of isoprene-degrading bacteria [17].

Isoprene-degrading bacteria are present in different environments, with fresh water, marine, and soils representing sinks for isoprene, with initial estimates suggesting that 20.4 Tg C yr^−1^ are consumed by soil [18,19]. Bacteria capable of growth on isoprene as the sole source of carbon and energy have been known for many years [20,21]. It was noted that two novel isoprene-degrading bacteria of the genera *Alcaligenes* and *Rhodococcus* were remarkably effective at biotransformations of trichloroethene and other halogenated hydrocarbons, although these organisms were not characterised in detail [21]. There is considerable conservation at the genetic level amongst isoprene degraders. All extant isoprene-degrading bacteria contain an isoprene degradation (*iso*) gene cluster, depicted in Figure 1. *isoA-F* encodes isoprene monooxygenase, a four-component soluble di-iron monooxygenase (SDIMO), which catalyses the epoxidation of isoprene to epoxyisoprene [22,23,24]. Isoprene monooxygenase (IsoMO) has been characterised in *Rhodococcus* sp. AD45, an actinobacterium isolated from freshwater sediment which has served as a model for isoprene metabolism research since the work of van Hylckama Vlieg et al. [22,23,24]. Epoxyisoprene is conjugated with glutathione by a glutathione *S*-transferase (IsoI), forming 1-hydroxy-2-glutathionyl-2-methyl-3-butene (HGMB), followed by two dehydrogenation steps catalysed by IsoH to form 2-glutathionyl-2-methyl-3-butenoate (GMBA) [22,23]. Further steps of isoprene degradation are still largely unknown, although it has been postulated that the glutathione moiety is removed and the product enters central metabolism through β-oxidation [24,25].

Recent studies have begun to reveal the diversity of isoprene-degrading bacteria in the environment, with soils, leaves, and estuarine sediments being targeted by cultivation-independent stable isoprene probing (SIP) to identify the active isoprene-utilising communities. The alpha-oxygenase component of isoprene monooxygenase, *isoA* (see below), has been identified as an excellent target for molecular probes when investigating the molecular ecology of isoprene degraders [26,27]. Actinobacteria are among the most commonly identified isoprene degraders, and members of the α, β, and γ-proteobacteria and the Bacteroidetes have also been identified [27,28,29,30]. *Rhodococcus* is often observed during studies of isoprene-degrading communities. Freshwater sediment enrichments contained approximately 50% relative abundance of *isoA* sequences related to *Rhodococcus*, while soil from beneath poplar, oil palm, and willow trees contained greater relative abundance of *isoA* relating to *Variovorax* spp., a member of the family Comamonadaceae (Betaproteobacteria) [26,27,28,30]. A recent study of willow soil reported that Comamonadaceae were enriched at 25 ppmv isoprene, a lower concentration than has previously been used in enrichment of isoprene degraders [30]. Members of the Comamonadaceae have also been identified following isoprene enrichment in soils from willow trees (*Salix spp*.) and a tyre dump, and also from the phyllosphere of poplar trees using DNA-stable isotope probing (DNA-SIP) methods, indicating that members of this family may be significant in microbial cycling of isoprene [26,27,28,30]. Isoprene-degrading bacteria from the α- and γ-Proteobacteria and Firmicutes have recently been isolated from the leaves and soil samples from tropical trees [31], and bacteria capable of co-oxidation of isoprene have also been isolated from soils, although none of these isolates have been characterised in detail [32].

*Variovorax* spp. are noted for their metabolic versatility [33,34], mutualistic interactions with plants and bacteria [35,36,37], and their distribution in varied ecological niches [38]. Members of this genus are noted for their ability to grow in environments contaminated by a range of aliphatic, aromatic and halogenated hydrocarbons [39,40,41]. The metabolic capabilities of *Variovorax* have been reviewed by Satola et al. [42]. Working with a novel *Variovorax* isolate, sp. WS11, this study aimed to confirm its mechanism of isoprene metabolism. We conducted a physiological characterisation to determine the similarities and differences in isoprene metabolism between *Rhodococcus* sp. AD45 and our Gram-negative isolate *Variovorax* sp. WS11. This study describes the genome of *Variovorax* sp. WS11 and reports on its functional potential, particularly relating to the abundance of various putative oxygenase genes. We link isoprene metabolism in *Variovorax* sp. WS11 to a conserved metabolic gene cluster, demonstrate the oxidative range of the IsoMO, and identify key functional differences between IsoMO and a well-characterised SDIMO from a methane-oxidising bacterium. *Variovorax* sp. WS11 has provided insights into a conserved mechanism of isoprene degradation and aid our understanding of the role of bacteria in the isoprene biogeochemical cycle. 

## 2. Materials and Methods 

### 2.1. Growth Conditions

*Variovorax* sp. WS11 [28] was grown in minimal medium (herein Ewers medium) [43] buffered to pH 6.0 with phosphate and supplemented with 1 mL/L vitamin solution [44]. Cultures were grown at 30 °C with shaking at 160 rpm. Unless stated otherwise, all cultures were grown in 120 mL vials sealed with butyl rubber stoppers. Typically, 1% *v*/*v* isoprene in the headspace of 120 mL vials was used. Headspace isoprene was measured according to Crombie et al. [45]. Sugars and carboxylic acids were used as growth substrates at 10 mM, and alkenes were used at 0.1% w/v or 1% *v*/*v* added to the headspace of 120 mL serum vials. Culture density measurements were recorded at 540 nm using a UV-1800 spectrophotometer (Shimadzu, Milton Keynes, UK).

### 2.2. Genome Sequencing, Annotation, and Analysis

DNA was extracted from Variovorax sp. WS11 cells grown using 10 mM glucose using a Genomic Tip 100/G DNA isolation kit (Qiagen, Manchester, UK) and corresponding DNA buffers (Qiagen) according to the manufacturer’s instructions. Genome sequencing was conducted by MicrobesNG (University of Birmingham, UK) using a combination of Illumina and Oxford Nanopore techniques. The final sequence, consisting of three contigs totalling 6.08 Mbp and two plasmids of 1.1 and 1.3 Mbp, was uploaded to MicroScope, https://mage.genoscope.cns.fr/microscope (accessed on 20/05/2019) [46]. Amino acid sequences of isoprene metabolism proteins of *Rhodococcus* sp. AD45 were used to query the *Variovorax* sp. WS11 genome by tBLASTn, https://blast.ncbi.nlm.nih.gov/Blast.cgi (accessed on 13/11/2019) [47] and located a single gene cluster on megaplasmid 1 (Figure 1). This *iso* gene cluster was visually inspected using Artemis software (release 16.0.0, Cambridge, UK) [48] and the individual *iso* genes queried against the NCBI protein database and against reported Iso amino acid sequences by BLASTp (Appendix A) [47].

### 2.3. Mutagenesis of isoA

*isoA* was deleted from *Variovorax* sp. WS11 by marker exchange mutagenesis according to Schäfer et al. [49]. Regions flanking *isoA* were synthesised by Q5 high-fidelity polymerase (NEB, Hitchin, UK) using primers *isoA_*up_F (5′-ATCAGGATCCCATGGGCCCGGACATCCCATTCG-3′)/*isoA*_up_R (5′-TGACTCTAGACGGCAGAAGCGGCTCTGCATCG-3′) and *isoA*_down_F (5′-ATCATCTAGAACATGGCGCAGACCTACC-3′)/*isoA*_down_R (5′-CTTAAAGCTTGATCGGCACCGTACACTC-3′) using *Variovorax* sp. WS11 genomic DNA as the template. Each region was cloned into pK18mobsacB and a gentamicin resistance cassette was excised from p34S-Gm [50] and ligated into the *XbaI* site between the *isoA* flanking regions. *Variovorax* sp. WS11 was prepared for electroporation according to Pehl et al. [51] with minor modifications. In brief, *Variovorax* sp. WS11 was grown in 50 mL culture using 10 mM glucose as the sole source of carbon and energy to an OD_540_ of 0.6, chilled at 4 °C for 48 h, then washed twice in chilled distilled water and finally resuspended in 1 mL of 10% (*v*/*v*) glycerol. Then, 100 μL aliquots were transformed by electroporation with 100 ng of the above construct, with pBBR1MCS-2 acting as a control for transformation efficiency. Exponential decay electroporation was used at 2.5 kV, 2.5 μF, 200 Ω in a 2 mm gap cuvette. Cells were allowed to recover for 3 h in 1 mL Ewers medium containing 10 mM glucose, pre-warmed to 30 °C for recovery, cultures were maintained at 30 °C and shaken at 160 rpm. Transformed cells were then plated onto Ewers agar containing 10 mM succinate and 50 μg/mL kanamycin, incubated at 30 °C for 1 week to select for single-crossover of the suicide construct into the genome. The pK18mobsacB backbone was removed from the genome by a second recombination event, selected for by spreading cells on Ewers agar containing 10 mM succinate, 10 μg/mL gentamicin and 5% sucrose (*w*/*v*), incubated at 30 °C for 5 days. Deletion of *isoA* was confirmed by loss of kanamycin resistance with concomitant maintenance of gentamicin resistance, as well as verification by PCR using primers *aacC1*_F (5′-TCGTGAGTTCGGAGACGTAG-3′)/*isoB*_R (5′-ACGTCGAAGCACTCCATCTC-3′).

### 2.4. Complementation of isoA Mutants

Function of the *iso* metabolic cluster was restored by complementing the mutant with pBBR:*isoAp*:*isoA-F* plasmid. The sequence including *isoA-F* with 256 bp of its native flanking region was generated by PCR using *isoAprom_fwd*(5′-ATCAGGTACCTTCCGACATCGATCGCGCAACC-3′) and *isoFprom_rev*(5′-TGATGGATCCCGATGGCGATCAGCTCGTAGTG-3′) under the assumption that the promoter would be located within the flanking region. The resulting product was digested with *KpnI* and *BamHI*. pBBR1MCS-2 [52] was digested with *KpnI* and *BamHI* and dephosphorylated using FastAP thermosensitive alkaline phosphatase (Thermo Scientific, Waltham, MA, USA). Both vector and digested PCR product were cleaned using a HighPure PCR product purification kit (Roche, Welwyn Garden City, UK) and ligated together with T4 DNA ligase (Thermo Scientific). The resulting mixture was used to transform *E. coli* Top10. Prepared plasmid DNA was sequenced and used to transform *Variovorax* sp. WS11 ∆*isoA* as described earlier. Colonies were grown in Ewers medium containing 10 mM succinate and then sub-cultured into Ewers medium containing 1% (*v*/*v*) isoprene.

### 2.5. Expression of iso Metabolic Genes in the Absence of Isoprene Monooxygenase

*Variovorax* sp. WS11 ∆*isoA* was grown to an OD_540_ of 0.6 in 50 mL Ewers medium containing 10 mM succinate. Cells were centrifuged at 7000× *g* for 10 min and washed in an equal volume of 50 mM HEPES, pH 6.0. Three replicate cultures were resuspended in 50 mL Ewers medium without a carbon source and starved for 2 h at 30 °C with shaking at 160 rpm. Three biological replicates were spiked with 10 mM succinate, 1% isoprene, or 0.1% (*v*/*v*) epoxyisoprene and incubated for a further 2 h before extracting RNA using a hot phenol extraction method [53]. Samples were treated with RNase-free DNase (Qiagen) to remove contaminating DNA and RNA was purified with an RNeasy minikit (Qiagen). RNA quality was checked by agarose gel electrophoresis. RNA concentration was determined using a NanoDrop spectrophotometer (Thermo Fisher) and DNA contamination was checked for by PCR using 16S rRNA primers. Superscript III reverse transcriptase (Invitrogen, Waltham, MA, USA) was used to synthesise complementary cDNA according to the manufacturer’s instructions using 500 ng template RNA, primed by random hexamers (Fermentas, Waltham, MA, USA). Negative controls were prepared in the absence of reverse transcriptase. Quantitative PCR was performed in 20 μL reactions using a StepOnePlus instrument (Applied Biosystems, Foster City, CA, USA) using Sensifast SYBR HI-ROX master mix (BioLine, London, UK) with 2 μL template cDNA. *isoG* was amplified by *qisoG_F* (5′-AAGACCATGAGCAACCAGGA-3′) and *qisoG_R* (5′-GCCGTTGTTCTCGACTTCAA-3′), and *rpoB* as a reference gene was amplified by q*rpoB_F* (5′-TGCAGGCCATTTACACCAAC-3′), q*rpoB_R* (5′-TTGAACTTCATGCGACCGAC-3′). Changes in *iso* gene expression were calculated relative to the *rpoB* reference gene according to the comparative C_T_ method, and further analysed by comparison of normalised C_T_ values between each carbon source [54].

### 2.6. Differential Expression of iso Metabolic Genes under Different Growth Conditions

*Variovorax* sp. WS11 was grown in triplicate in 20 mL Ewers medium containing 1% (*v*/*v*) isoprene, 10 mM glucose or 10 mM glucose + 1% isoprene. Cells were harvested by centrifugation at 7000× *g* for 10 min and resuspended to an OD_540_ of 10.0 in 1 mL HEPES, pH 6.0. Cell suspensions were sealed in 30 mL vials with butyl rubber stoppers and pre-heated to 30 °C in a water bath, shaken at 160 rpm for 3 min before adding 300 ppmv isoprene. After incubation for 1 min, 50 μL of headspace gas was sampled using a gas-tight syringe (Agilent, Cheadle, UK) and isoprene was measured using a Fast Isoprene Sensor (Hills-Scientific, Boulder, CO, USA) at 3 min intervals for 1 h. Depletion of isoprene was used to calculate the rate of isoprene uptake by cells grown under each condition.

*Variovorax* sp. WS11 was prepared for RNA extraction by growing in 50 mL Ewers medium in 120 mL vials using the same substrate combinations described above. Cells were harvested at an OD_540_ of 0.6 by centrifugation at 7000× *g* for 10 min, then total RNA was extracted as described above. *isoA* was amplified using q*isoA_F* (5′-GATGTCTCGTTCTGGCGTTC-3′), q*isoA_R* (5′-ACCCGTAGTCCTTCATCGTG-3′). Expression of *isoA* under each growth condition was calculated relative to expression of *rpoB*.

### 2.7. Oxidation of Alkenes by Variovorax sp. WS11 Grown under Different Conditions

*Variovorax* sp. WS11 was grown in 2 litre flasks in 400 mL Ewers medium containing either 10 mM glucose or 10 mM succinate as the sole source of carbon and energy. Cells were grown to an OD_540_ of 0.6, harvested by centrifugation at 7000× *g* for 10 min and resuspended in 4 mL of 50 mM HEPES, pH 6.0. Cells were drop frozen in liquid nitrogen and stored at −80 °C until required. *Variovorax* sp. WS11 was grown in 4 litres of Ewers medium using isoprene as the sole source of carbon and energy in a 4 litre working volume fermentor (Electrolab, Tewkesbury, UK) with constant maintenance of optimal growth conditions (30 °C, 160 rpm, pH 6.0) and air flow (2.4 L/min). Isoprene was supplied by bubbling air (1 mL min^−1^) through a small volume of liquid isoprene contained in a 30 mL vial, held at 0 °C on ice (Appendix A). Purity was routinely checked by a combination of microscopy and growth on R2A agar (Oxoid, Basingstoke, UK). Cells were harvested every 48 h and drop frozen under the conditions described above. Substrate oxidation rates were tested for cells prepared under each condition using a Clark oxygen electrode [55]. Substrate-induced rates of oxygen uptake were calculated according to the previously described method [56]. Drop-frozen *Variovorax* sp. WS11 cells from each of the above growth conditions were thawed, starved at room temperature for 2 h, and resuspended in 3 mL of 50 mM HEPES, pH 6.0, to an OD_540_ of 2.0 and warmed to 30 °C with stirring. Oxidation substrates were added to 100 μM as saturated solutions in water. Endogenous rates of oxygen uptake were subtracted from oxygen uptake after substrate addition to calculate substrate-induced rates of oxygen uptake for each growth condition using three biological replicates.

### 2.8. Inhibition of IsoMO and sMMO Activity by Alkynes

C2–C4 alkynes were added from gaseous stocks. Gas–liquid partitioning was calculated according to the Henry’s Law constants (*H^cp^*) of acetylene (4.1 × 10^−4^ mol/m^3^ Pa), propyne (9.3 × 10^−4^ mol/m^3^ Pa), and butyne (5.4 × 10^−4^ mol/m^3^ Pa) [57] to enable addition of each alkyne to 50 μM. C6–C8 alkyne stock solutions were prepared as 1% (*v*/*v*) in dimethyl sulfoxide (DMSO). *Variovorax* sp. WS11 was used for isoprene uptake assays at an OD_540_ of 1.0 in 1 mL HEPES (50 mM, pH 6). Isoprene-grown cells were placed in 30 mL serum vials sealed with butyl rubber stoppers and pre-warmed for 3 min at 30 °C, with shaking at 160 rpm. The 300 ppm isoprene was added and *Variovorax* sp. WS11 was incubated for a further 1 min before sampling 50 μL headspace as previously described. Samples were taken at 5 min intervals for 45 min. Alkynes were added (to an aqueous concentration of 50 μM) as gases to the headspace or as concentrated stock in DMSO immediately after sampling timepoint 3. The inhibited rate (timepoints 5–10) was calculated relative to the initial rate of isoprene oxidation (timepoints 1–3). Samples without alkyne and with DMSO alone were used as controls. Each assay was performed using three biological replicates. With the exception of C6–C8 alkynes, results are reported as the mean of these three replicates ± standard deviation. C6–C8 alkynes were presented as the mean inhibition by DMSO (1.4 ± 7.2%) subtracted from the mean inhibition by alkyne ± standard deviation about the means.

*Methylococcus capsulatus* (Bath) was grown in a 2 litre fermentor (New Brunswick, Eppendorf, Stevenage, UK) using methane as the sole source of carbon and energy as previously described [58], with the exception that this culture was grown in batch mode. The switch from expression of particulate to soluble methane monooxygenase was monitored using the naphthalene assay [59]. Cells were drop frozen in liquid nitrogen and stored at −80 °C. When thawed, cells were resuspended in 1 mL of 50 mM phosphate, and pH 7.0 20 mM sodium formate was included as a source of reducing power. Isoprene uptake assays were performed as described above, with cells pre-heated to 30 °C and the rate of isoprene uptake calculated over 45 min. As a proxy for the rate of methane oxidation by sMMO, the conversion of propylene to propylene oxide was used as a measure of typical sMMO activity, with propylene oxide formation detected by gas chromatography as previously described [60]. Alkyne inhibition of isoprene or propylene oxidation by sMMO was tested by the above method using 50 µM acetylene or 50 µM octyne. Since DMSO alone also had an inhibitory effect on the sMMO (but not on IsoMO) and octyne was added as a solution in DMSO, the inhibition of propylene oxidation and isoprene oxidation attributable to DMSO, 25.2 ± 6% and 27.4 ± 5% respectively, was subtracted from the inhibition of substrate oxidation due to octyne.

### 2.9. Heterologous Expression of Isoprene Monooxygenase

*isoA-F* was amplified by Q5 high-fidelity polymerase using primers *isoA_fwd* (5′-ATGAACATGTCCTTGCTGAGCCGAGACGACTG-3′) and *isoF_rev* (5′-ATCAGGATCCCGATGGCGATCAGCTCGTAG-3′). The purified PCR product was digested with *BamHI* and *PciI* and ligated with pTipQC1 [61] that had been digested with *BamHI* and *NcoI*. The resulting vector (pTipQC1:iso11) was used to transform *Rhodococcus* sp. AD45-id [28] by electroporation as previously described [28]. IsoMO expression was induced as previously described [28]. In brief, *Rhodococcus* sp. AD45-id containing either pTipQC1 or pTipQC1:iso11 was grown in 120 mL serum vials sealed with butyl rubber stoppers, in 20 mL Ewers medium containing 10 mM succinate to an OD_540_ of 0.6 and induced with 1 µg/mL thiostrepton (prepared as 20 mg/mL stock in DMSO). Induced cells were incubated for 16 h at 30 °C with shaking at 160 rpm and then centrifuged at 7000× *g* for 10 min. Pellets were resuspended to an OD_540_ of 10.0 in 50 mM phosphate buffer (pH 7.0). Cells were tested for isoprene consumption using the Fast Isoprene Sensor as described earlier. Three biological replicates were used for each assay.

### 2.10. Accession Number

The chromosomal and megaplasmid sequences of *Variovorax* sp. WS11 have been deposited at DDBJ/ENA/Genbank under the accession JAAGOW000000000. The version described in this paper is version JAAGOW010000000. The genomic nucleotide sequence of *Ramlibacter* sp. WS9 is available under accession number NZ_RKMB00000000.

## 3. Results and Discussion

### 3.1. Isolation of Variovorax sp. WS11 from Willow Soil

*Variovorax* sp. WS11 was previously isolated from willow soil samples by continuously enriching in the presence of 25 ppmv isoprene, as a comparatively low concentration [28]. *Variovorax* sp. WS11 was one of two Gram-negative isoprene-degrading bacteria isolated from this environment, together with *Ramlibacter* sp. WS9 [30], and was shown by 16S rRNA gene sequencing to be a novel species of *Variovorax*. Phylogenetic analysis of the 16S rRNA gene of *Variovorax* sp. WS11 indicates that it is most related to *Variovorax* sp. RA8 (Appendix A), with 16S rRNA nucleotide sequence identity of 99.9% with 99% query cover [62]. *Variovorax* sp. RA8 was isolated from Japanese river sediment and has been noted for its ability to degrade the herbicide linuron [63].

### 3.2. Genome sequencing and Analysis

The genome of *Variovorax* sp. WS11 was sequenced by a combination of Illumina and Oxford Nanopore techniques and annotated using the MicroScope Microbial Genome Annotation and Analysis Platform [46]. Two megaplasmids of 1.1 and 1.3 Mbp were identified (Appendix A). The sequence of the chromosome was comprised of three contigs totalling 6.08 Mbp. Relatively little pseudogenisation was detected, with only 26 pseudogenes detected by the MicroScope annotation pipeline [46]. The GC content of the *Variovorax* sp. WS11 chromosome was higher than that of both megaplasmids, measuring 68.2% compared to 64.4% and 66.6% for megaplasmids 1 and 2, respectively (Appendix A). Of 5930 predicted coding sequences, 1972 were not assigned a function by the RAST annotation pipeline, http://rast.theseed.org/FIG/rast.cgi (accessed on 16/05/2019) [64,65,66]. RAST subsystems analysis predicted that megaplasmids 1 and 2 dedicate a large portion of their coding sequences to metabolic processes (Appendix A). While 2.1% of the chromosomal coding sequences had predicted functions relating to fatty acids, lipids and isoprenoids, megaplasmids 1 and 2 had 5.8% and 5.7%, respectively, of their total coding sequences dedicated to this subsystem. In total, 4.6% of chromosomal genes and 4.5% of megaplasmid 1 genes were predicted to encode functions relating to carbohydrate metabolism, while 7.5% of megaplasmid 2 genes related to this subsystem (Appendix A). Megaplasmid 2 was also predicted to encode asparaginyl-tRNA, while all other genes involved in tRNA biosynthesis are located on the chromosome.

#### 3.2.1. The Genome of *Variovorax* sp. WS11 Encodes Multiple Putative Oxygenase Gene Clusters

A combination of visual inspection, tBLASTn analysis using reference genes [47], and annotation by the MicroScope pipeline [46] identified an abundance of partial and apparently complete oxygenase gene clusters in the genome of *Variovorax* sp. WS11. One such gene cluster was predicted to encode a methanesulfonate monooxygenase (*msmABCD*), located on the chromosome, which resembled the well-characterised *msm* cluster of *Methylosulfonomonas methylovora* [67]. The case for this putative gene cluster encoding an active methanesulfonate monooxygenase (MsaMO) was strengthened by the presence of a characteristic Rieske-type [2Fe-2S] motif (CXH-X_26_-CXXH) within *msmA* [67]. *msmB*, predicted to encode the hydroxylase β-subunit, was located immediately downstream of *msmA* (Appendix A). Likewise, *msmC* and *msmD*, encoding the ferredoxin and reductase components respectively, were positioned downstream of *msmAB*. However, the putative MsaMO gene cluster in *Variovorax* sp. WS11 lacked the typical transporter genes, *msmEFGH*, expected to be in close proximity to *msmABCD* [68]. Instead, a putative proline/betaine transporter and a putative sulphite exporter were located downstream of *msmD*. Homologs of *msmEFGH* were detected elsewhere on the genome by tBLASTn analysis using the genes of *M. methylovora* as queries. *Variovorax* sp. WS11 was inoculated in Ewers medium with 5 mM sodium methanesulfonate, or 5 mM ethanesulfonic acid but no growth was observed, confirming that the putative MsaMO genes alone were insufficient to support growth under the conditions tested. Secondly, several *tauD* genes were identified in both the chromosome and megaplasmid 2 of *Variovorax* sp. WS11, predicted to encode α-ketoglutarate-dependent taurine dioxygenase (Appendix A). While none of these were in proximity to the corresponding *tauABC* genes which encode an ABC-type transporter required for uptake of taurine into the cell [69], *tauABC* were identified elsewhere on megaplasmid 2. *Variovorax* sp. WS11 was capable of utilising taurine as a sole source of carbon and energy (Appendix A), indicating that an active taurine dioxygenase (TauD) was expressed or that an alternative oxygenase capable of taurine metabolism was expressed. Two other oxygenase gene clusters were identified, each predicted to encode a salicylate 5-hydroxylase (*nagGH*) (Appendix A). Each putative salicylate 5-hydroxylase gene cluster had differences in organisation. Both were located on megaplasmid 2 but in opposing orientations and distant from each other, and each contained a copy of a putative NAD(P)-dependent reductase (*nagAa*) and a ferredoxin (*nagAb*) (Appendix A). Salicylate 5-hydroxylase is often utilised as part of the wider naphthalene degradation pathway, with naphthalene oxidation by naphthalene 1,2-dioxygenase (*nagAcAd*) to salicylate comprising the upper part of the pathway [70]. The lack of genes involved in the subsequent salicylate oxidation (*nahAa-nahAD*) [71] in the genome of *Variovorax* sp. WS11 made it unlikely for growth on salicylate or naphthalene to occur, and neither 5 mM naphthalene or 5 mM sodium salicylate supported growth of *Variovorax* sp. WS11 (Appendix A).

#### 3.2.2. The *iso* Metabolic Gene Cluster of *Variovorax* sp. WS11 is Distinct from that of *Rhodococcus* sp. AD45

Using the translated amino acid sequences of the *iso* metabolic genes of *Rhodococcus* sp. AD45 as tBLASTn query sequences, putative *iso* metabolic genes were located on a single gene cluster on megaplasmid 1 in *Variovorax* sp. WS11 (Figure 1, Appendix A) [47]. To the best of our knowledge, this is the first isolated species of *Variovorax* which contains the full *iso* metabolic gene cluster, as determined by tBLASTn analysis of the critical *isoA* gene against the NCBI protein database [47]. Visual inspection using Artemis software revealed a similar organisation of *iso* metabolic genes in *Variovorax* sp. WS11 as found in *Rhodococcus* sp. AD45 [48]. Aside from *isoA*, the amino acid identities of translated *isoB-J* genes of *Variovorax* sp. WS11 and *Rhodococcus* sp. AD45 shared homology ranging from approximately 40–60% (Appendix A). tBLASTn analysis was conducted using the translated *iso* amino acid sequences of *Ramlibacter* sp. WS9, a second Gram-negative isoprene degrader isolated from willow soil [30]. Alignment of amino acid sequences of *isoA* of known isoprene-degrading bacteria with reference oxygenase alpha subunits of other SDIMO reveals an interesting evolutionary relationship (Figure 2). A number of Gram-positive actinobacteria have been isolated with the ability to degrade isoprene, and the *isoA* sequences of these organisms are typically grouped together when phylogenetically analysed. *Variovorax* sp. WS11 and *Ramlibacter* sp. WS9, the first reported Gram-negative isoprene degraders from soil, occupy a distinct branch of the phylogenetic tree, more closely related to the alkene monooxygenase alpha subunit *xamoA* of *Xanthobacter* strain Py2. The *isoA* sequences of *Variovorax* sp. WS11 and *Ramlibacter* sp. WS9 share 92% amino acid identity, while the former only shares 73% amino acid identity with the *isoA* sequence of *Rhodococcus* sp. AD45 (Appendix A). Enzymes involved in subsequent steps in isoprene metabolism following the conversion of isoprene to epoxyisoprene by IsoMO, encoded by *isoGHIJ*, are duplicated in *Rhodococcus* sp. AD45 [45]. No duplication of *isoGHIJ* was found in *Variovorax* sp. WS11 (Figure 1). This also appears to the be case for other Gram-negative isoprene-degrading bacteria, including *Ramlibacter* sp. WS9 (Figure 1), and for a few Gram-positive isoprene-degrading bacteria [28].

Interestingly, *Variovorax* sp. WS11 lacks glutathione biosynthesis genes in the isoprene gene cluster. Glutathione is important in many bacteria as a means of protection against toxic and oxidative stresses during aerobic growth, but Gram-positive bacteria typically use alternative low molecular weight thiols such as mycothiol, and do not produce glutathione [72]. Despite this, previous work strongly indicated the importance of glutathione in isoprene degradation by *Rhodococcus* sp. AD45, since IsoI detoxifies the epoxyisoprene produced from isoprene by IsoMO by producing a glutathione conjugate, 1-hydroxy-2-glutathionyl-2-methyl-3-butene (HGMB) [22,23,24]. The styrene oxidation pathway of the Gram-positive strain *Gordonia rubripertincta* CWB2 also involves glutathione [73]. In common with other Gram-negative bacteria, *Variovorax spp.* produce glutathione as a core metabolic process, and all predicted glutathione biosynthesis genes are located on the chromosome. It was predicted that *Rhodococcus* sp. AD45 gained the genes required for glutathione biosynthesis (*gshAB*), located in the *iso* metabolic gene cluster (Figure 1), by horizontal gene transfer [74]. Further differences in the organisation of the *iso* metabolic gene cluster include the insertion of a single copy of *aldH* between *isoJ* and *isoA* in *Variovorax* sp. WS11, a trait conserved in *Ramlibacter* sp. WS9 [30], while *Rhodococcus* sp. AD45 contains two distinct *aldH* genes located at opposite ends of the *iso* gene cluster. The single *aldH* gene in the *iso* metabolic gene cluster of *Variovorax* sp. WS11 is most closely related to *aldH1* in *Rhodococcus* sp. AD45, with 52.7% amino acid identity (97% coverage) compared to only 26.1% amino acid identity with *aldH2* (69% coverage), predicted to encode a 4-hydroxymuconic semialdehyde dehydrogenase [47,75]. The roles of *aldH1 and aldH2* in isoprene metabolism are currently unconfirmed, although the *tadD* gene of *Rhodococcus* sp. PD630, which shares 81% amino acid identity with *aldH1* of *Rhodococcus* sp. AD45 [45], encodes a glyceraldehyde 3-phosphate dehydrogenase [76]. *Rhodococcus* sp. AD45 contains two putative regulatory genes annotated as *marR*, with only the divergently transcribed *marR2* being upregulated in response to isoprene [45]. The closest *marR* homolog in *Variovorax* sp. WS11 also resides on megaplasmid 1 but is 200 kbp in distance from the *iso* gene cluster. Instead, two putative LysR-type transcriptional regulators, annotated as *dmlR*, are divergently transcribed from the *iso* cluster in *Variovorax* sp. WS11 (Figure 1), suggesting that this novel isolate employs an alternative method of regulation of isoprene metabolism. The *iso* metabolic gene cluster of *Ramlibacter* sp. WS9 also contains a single copy of *dmlR* upstream of *isoG*, potentially indicating that these closely related isoprene degraders share a similar method of regulating the expression of *iso* metabolic genes. *Variovorax* sp. WS11 was selected as a Gram-negative model for the study of isoprene metabolism primarily due to two factors. The growth rate of *Variovorax* sp. WS11 on isoprene is greater than that of *Ramlibacter* sp. WS9, measuring 0.052 ± 0.004 h^−1^ [28]. Secondly, as genetics systems have previously been established for *Variovorax paradoxus* EPS [51], we set out to optimise these techniques to allow the manipulation of *Variovorax* sp. WS11.

**Figure 2 microorganisms-08-00349-f002:**
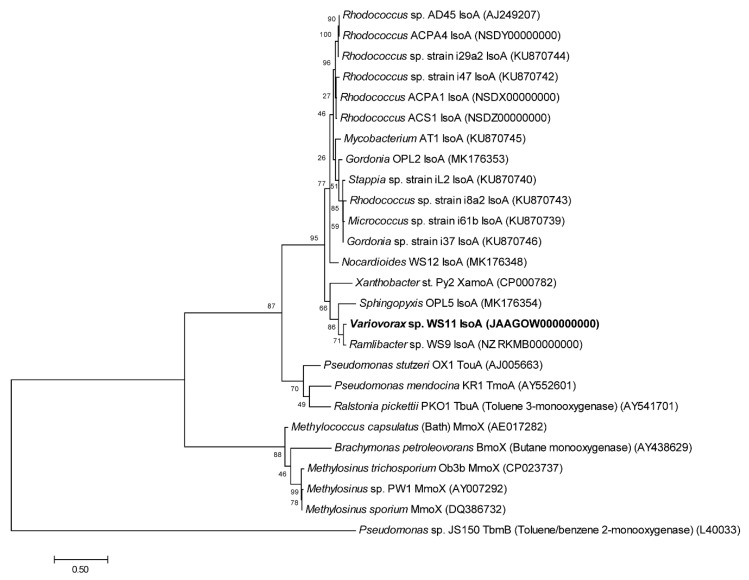
Phylogenetic organisation of isoprene monooxygenase oxygenase alpha-subunit (*isoA*) against reference soluble di-iron monooxygenase (SDIMO) amino acid sequences with accession numbers in brackets. The tree was drawn using the Maximum Likelihood method using Mega7 [77]. Bootstrap values (500 replications) are shown at the nodes. All positions containing gaps and missing data were eliminated. Branch lengths are measured in the number of substitutions per site.

### 3.3. Functional Confirmation and Substrate-Induced Transcription of iso Metabolic Genes

Functional confirmation of putative isoprene monooxygenase-encoding genes has previously been achieved by expression of *isoA-F* in *Rhodococcus* sp. AD45-id, a strain cured of the megaplasmid containing the isoprene gene cluster, which is therefore unable to metabolise isoprene [28]. The IsoMO genes (*isoA-F*) from *Variovorax* sp. WS11 were cloned into the *Rhodococcus* expression vector pTipQC1 [61], under the control of a thiostrepton-inducible promoter. Whole-cells of *Rhodococcus* sp. AD45-id containing pTipQC1:iso11 were capable of oxidising isoprene at 38% of the wild-type rate when induced, in contrast to cells transformed with pTipQC1 without an *isoA-F* insert (data not shown), confirming that the *iso* metabolic gene cluster identified in *Variovorax* sp. WS11 was active. 

To establish beyond doubt that the isoprene monooxygenase was required for isoprene metabolism in *Variovorax* sp. WS11, targeted mutagenesis was used. Deletion of *isoA* abolished all growth on isoprene (Figure 3), but growth on glucose and succinate remained similar to that of the wildtype (data not shown). The mutant strain *Variovorax* sp. WS11 ∆*isoA* could be complemented for growth on isoprene by expression of *isoA-F* including 256 nucleotides 5’ of *isoA*, containing the full intergene region, from pBBR1MCS-2, restoring growth on isoprene to approximately wild-type levels (Figure 3). Since *isoGHIJ* encode the subsequent enzymes of the pathway, including those required for detoxification of epoxyisoprene, it was anticipated that the whole cluster *isoGHIJABCDEF* would be co-induced during conditions which activated isoprene metabolism. The first gene of the cluster, *isoG*, was selected to determine the levels of gene transcription in response to the potential inducers isoprene and epoxyisoprene. Gene expression was studied in *Variovorax* sp. WS11 ∆*isoA* as this strain is unable to metabolise isoprene, meaning that any upregulation in the presence of isoprene could not be explained by any regulatory effects of subsequent products of isoprene metabolism. RT-qPCR analysis revealed that, in WS11 ∆*isoA*, *isoG* was significantly upregulated by both isoprene (6.3-fold) and epoxyisoprene (146-fold) after 2 h of exposure to the inducing substrate (Appendix A). This contrasts with our knowledge of *Rhodococcus* sp. AD45, in which epoxyisoprene induced gene expression, but isoprene itself had no inducing effect [45]. Given the significant increase in upregulation by epoxyisoprene compared to isoprene, it is likely that epoxyisoprene has a much greater affinity for regulatory elements specific to the *iso* metabolic gene cluster, although downstream products of epoxyisoprene metabolism, such as HGMB or GMBA or an as-yet unidentified intermediate, may act as the primary inducer of *iso* gene expression in *Variovorax* sp. WS11.

### 3.4. Differential Expression of a Broad-Range SDIMO Active with Branched Hydrocarbons

Growth of *Variovorax* sp. WS11 was sustained by a range of sugars, carboxylic acids and amino acids, but not by any alkenes tested with the exception of isoprene (Appendix A). When *Variovorax* sp. WS11 was grown solely on isoprene, whole cells oxidised 1.51 ± 0.64 nmoL isoprene/min/mg, but when grown on glucose or a combination of glucose and isoprene, whole cells did not oxidise isoprene. RT-qPCR analysis of *isoA* expression under the same growth conditions confirmed that *isoA* was upregulated significantly when *Variovorax* sp. WS11 was grown on isoprene, with a relative expression value of 582 when compared to isoA expression by cells grown using glucose, but expression of *isoA* was repressed during growth on glucose or a combination of glucose and isoprene (Appendix A). 

SDIMOs are typically capable of oxidising a wide range of organic substrates. The substrate specificity of IsoMO was investigated by comparing the substrate oxidation profile of *Variovorax* sp. WS11 when grown on glucose, succinate or isoprene. Cells grown on succinate and glucose were only able to oxidise sugars and carboxylic acids, while isoprene-grown cells were also able to oxidise a wide range of alkenes (Figure 4). The substrate oxidation profile of *Variovorax* sp. WS11 indicated a positive correlation between rate of oxidation and carbon chain length, with octene and dodecene being oxidised at very high rates compared to ethylene and propylene (Figure 4). A consistent trend is seen when comparing oxidation of branched and unbranched hydrocarbons; isoprene (2-methyl-1,3-butadiene) and 3-methyl-1,4-pentadiene are each oxidised at higher rates than 1,3-butadiene and 1,4-pentadiene, respectively. It is also curious that IsoMO is incapable of oxidising benzene but all branched derivatives (ethylbenzene, propylbenzene, toluene, *o*-xylene, styrene) are oxidised. The same is observed for cyclohexene and methyl-cyclohexene, with activity only being observed in the branched derivative. 

### 3.5. Inhibition by Alkynes Indicates that IsoMO is a Distinct SDIMO

Alkynes have been known to act as inhibitors of SDIMO activity for many years. Methane oxidation by the soluble methane monooxygenase (sMMO) of *Methylococcus capsulatus* (Bath) was completely inhibited by relatively low concentrations of acetylene [78,79], and this inhibition was dependent on turnover of acetylene at the active site [79]. It was previously observed that the rate of substrate oxidation by the sMMO of *M. capsulatus* (Bath) decreased with increasing carbon chain length [60]. Therefore, due to the decreased rate of substrate turnover, it may be the case that longer-chain alkynes exert a weaker inhibitory effect on sMMO activity than acetylene.

#### 3.5.1. Inhibition of IsoMO Correlates with Increasing Alkyne Chain Length

Alkynes (C2–C8) were tested for their ability to inhibit isoprene oxidation by IsoMO in *Variovorax* sp. WS11. The rate of isoprene oxidation was decreased by 6.5% by 50 μM acetylene, demonstrating that acetylene is a significantly less potent inhibitor of IsoMO than of sMMO (Figure 5). Inhibition increased significantly with increasing chain length; propyne and butyne inhibited 53.1 and 87.4% of isoprene oxidation activity, respectively. Hexyne and heptyne each inhibited to a similar degree as butyne, with 83.3 and 88.2% inhibition, respectively, while octyne was the most potent inhibitor tested, causing 94.7% inhibition of isoprene oxidation under the conditions tested (Figure 5). DMSO had no inhibitory effect on isoprene oxidation by IsoMO, validating the observed inhibition of IsoMO by C6–C8 alkynes.

#### 3.5.2. IsoMO is Distinct from the Well-Characterised sMMO

The sMMO from *Methylococcus capsulatus* (Bath) was selected as a comparative SDIMO for inhibition by alkynes due to the pre-existing knowledge of the potency of acetylene as an inhibitor of this sMMO [78,79], and since the sMMO of *Methylocella silvestris* BL2 has previously been reported to co-oxidise isoprene [80], although neither of these methanotrophs can grow on isoprene. Therefore, the sMMO of *M. capsulatus* (Bath) was tested and confirmed to co-oxidise isoprene with an initial rate of 1.38 ± 0.15 nmoL/min/mg dry weight. A total of 50 μM acetylene and 50 μM octyne were the least potent and most potent inhibitors (respectively) of isoprene oxidation by the IsoMO of *Variovorax* sp. WS11 under the conditions tested (Figure 5) and were also tested as inhibitors of isoprene oxidation by sMMO. The assay conditions were first verified by testing the co-oxidation of propylene to propylene oxide by sMMO. On average, 96.2% of propylene oxidation was inhibited by 50 μM acetylene, confirming the efficacy of this inhibitor and immediately differentiating IsoMO from sMMO, compared to 99.1% of isoprene oxidation by sMMO by 50 μM acetylene, 15-times its inhibition of isoprene oxidation by IsoMO. 50 μM octyne did not inhibit propylene oxidation by sMMO under these conditions, but approximately 5.7% of isoprene oxidation was inhibited by octyne (Figure 5). These data demonstrate the effectiveness of these alkyne inhibitors to distinguish between true isoprene degraders and bacteria which co-oxidise isoprene using an alternative SDIMO.

## 4. Conclusions

This study describes the genetic and biochemical characterisation of a new model bacterium for isoprene degradation, the Gram-negative bacterium *Variovorax* sp. WS11. The genetic tractability of this model organism was demonstrated, and the use of a Gram-negative isolate may aid in the functional characterisation of the downstream steps of the isoprene metabolic pathway, such as the function of *isoG, isoJ,* and *aldH*. The *iso* metabolic gene cluster in *Variovorax* sp. WS11 was located on a megaplasmid, suggesting that this second well-characterised *iso* gene cluster was acquired through horizontal gene transfer. The *iso* metabolic gene cluster displayed significant variation from the original model of *Rhodococcus* sp. AD45, such as the lack of duplications of *isoGHIJ* and *aldH*, and the lack of glutathione biosynthesis genes. Functionality of *isoA-F* was confirmed by expression in *Rhodococcus* sp. AD45-id, and the dependency of *Variovorax* sp. WS11 on *iso* genes was confirmed by knockout of *isoA*. Two putative LysR-type transcriptional regulators may play a role in regulating isoprene metabolism and remain the subjects of further study. IsoMO oxidised a wide range of alkenes, with a tendency to display increased activity with branched hydrocarbon chains when compared to their unbranched counterparts. Inhibition of isoprene oxidation by 50 μM acetylene and 50 μM octyne differed significantly between IsoMO and sMMO. An estimated 2.4 million tons of isoprene are degraded each year by soils [18,19]. The proportion of isoprene uptake by bona fide isoprene degraders compared to bacteria which co-oxidise isoprene using other SDIMOs is currently unknown. Future experiments using soil samples can now be carried out to determine the efficacy of this test in situ with environmental samples, with a view to quantifying the contributions of different groups of SDIMO-utilising organisms in the removal of isoprene from the biosphere. This study provides a basis for the continued investigations into the role of bacteria in the biogeochemical cycling of isoprene using a comparative Gram-negative model isoprene degrader to study isoprene metabolism.

## Figures and Tables

**Figure 1 microorganisms-08-00349-f001:**
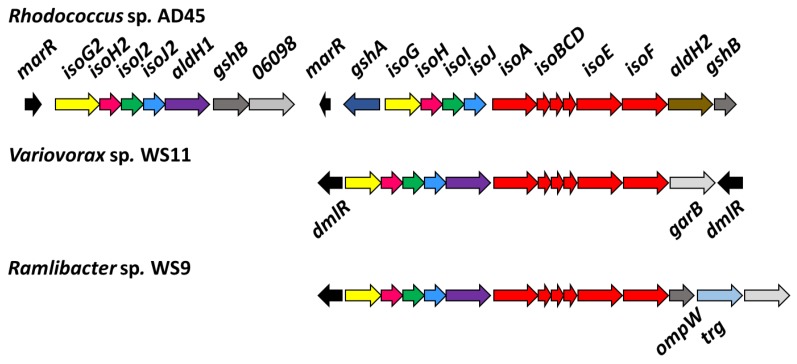
Comparison of *iso* metabolic gene cluster organisation of the reference isoprene degrader *Rhodococcus* sp. AD45 with *Variovorax* sp. WS11 and *Ramlibacter* sp. WS9.

**Figure 3 microorganisms-08-00349-f003:**
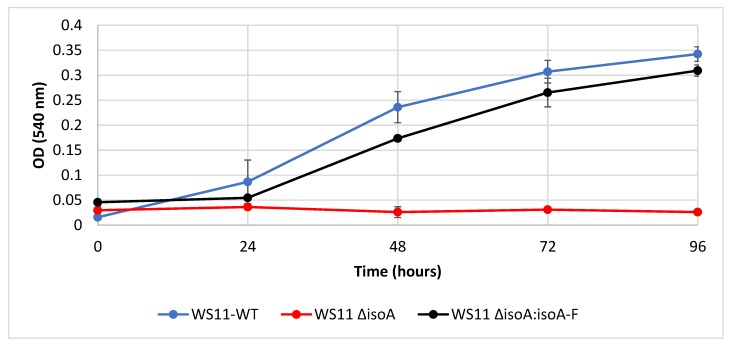
Wild-type *Variovorax* sp. WS11 (WS11-WT) and complemented *isoA* mutant (WS11 ∆*isoA*:*isoA-F*) grow to a similar culture density when provided with 1% *v*/*v* isoprene, but *isoA* deletion strain (WS11 ∆*isoA*) cannot grow on isoprene. Time points are the mean of three biological replicates and the error bars represent the standard deviation.

**Figure 4 microorganisms-08-00349-f004:**
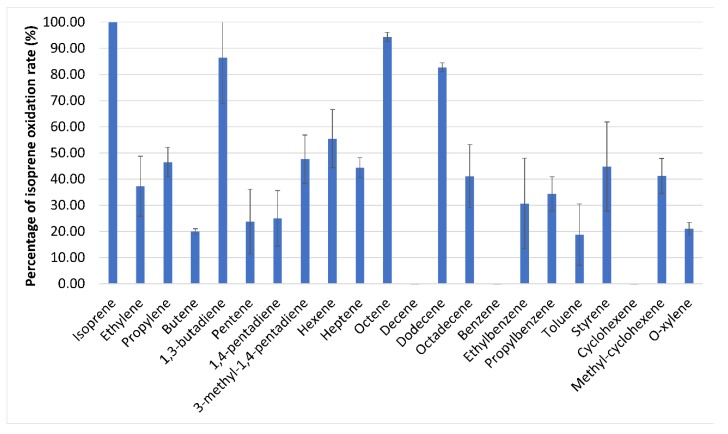
Relative rates of substrate-dependent oxygen uptake by isoprene-grown *Variovorax* sp. WS11, where 100% is 31.2 nmoL/min/mg dry weight.

**Figure 5 microorganisms-08-00349-f005:**
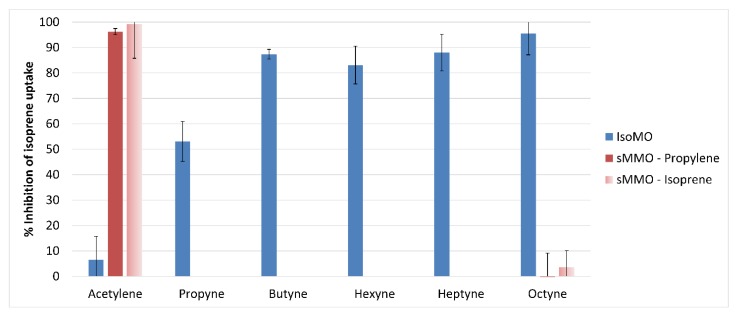
Differential inhibition of IsoMO and soluble methane monooxygenase (sMMO) by C2–C8 alkynes (50 µM), where DMSO acts as a control for C6–C8 alkynes since C6–C8 alkynes were dissolved in DMSO before addition.

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
