# Peer review of "Isoprene Oxidation by the Gram-Negative Model bacterium Variovorax sp. WS11"

_microorganisms, 2020, doi:10.3390/microorganisms8030349_

Round 1

Reviewer 1 Report

In the article authors report the characterization of a novel Gram-negative bacterium, Variovorax sp. WS11, which degrades isoprene, a plant-producing secondary metabolite which contributes to the global warming effect. The manuscript is well written and can be published in Microorganisms journal. This work went from the well-known lab of Prof. C.Murrell, famous for their research in C1-compounds degrading microbes and enzymes. The enzyme for isoprene degradation was heterolgically expressed in rhodobacter cells and its activity was measured toward isoprene oxidation. All experiments were well planed and clearly explained, Results and Discussion section is satisfactory explain the results obtained and compare it with previously described for that in Rhodococcus sp. AD45 cells. This article will be of importance for the understanding the pathways of isoprene metabolism in Nature.

Author Response

Dear Reviewer

Thank you for your very kind and supportive comments for the manuscript “Isoprene oxidation by the Gram negative model bacterium Variovorax sp. WS11”. Your comments have been greatly appreciated.

Yours sincerely,

Robin Dawson

Reviewer 2 Report

The manuscript describes the genetic inventory for isoprene degradation in  Variovorax sp., a gram-negative bacterium. The functionality of the genes was confirmed by mutagenesis, complementation and biochemical characterizations.

It is a very solid study. Below are minor comments (mostly editorial) changes, which can further improve the result description.

P3L124-126. A duplicating sentence. 

P4L162. Indicate the source of pBBR plasmid.  Provide description for pBB1MCS-2 plasmid (described on P10L420); Indicate how the promoter region was identified.

Figure 2. Highlight proteins from Variovorax; Add bootstrap values.

P10L411. I did not see methods for pTipQC1 plasmid construction.

Author Response

Dear Reviewer

Thank you for your very kind and supportive comments on the manuscript “Isoprene oxidation by the Gram-negative model bacterium Variovorax sp. WS11”. Your comments have been addressed below.

P3L124-125. A duplicating sentence. 

The duplicated sentence identified by Reviewer 2 has been deleted. The sentence now reads “DNA was extracted from Variovorax sp. WS11 cells grown using 10 mM glucose using a Genomic Tip 100/G DNA isolation kit (Qiagen) and corresponding DNA buffers (Qiagen) according to the manufacturer’s instructions.

P4L162. Indicate the source of pBBR plasmid.  Provide description for pBB1MCS-2 plasmid (described on P10L420); Indicate how the promoter region was identified.

The original study describing the construction of pBBR1MCS-2 [Kovach et al., 1995] has now been cited on P4L166. A description of the identification of the promoter region upstream of isoA is included on P4L162, and P4L164-165.

Figure 2. Highlight proteins from Variovorax; Add bootstrap values.

Figure 2 has been edited accordingly. The IsoA sequence from Variovorax sp. WS11 is now highlighted and bootstrap values have now been included. P10L409-410 describes the bootstrap value (500 replications) for the phylogenetic tree.

P10L411. I did not see methods for pTipQC1 plasmid construction.

These methods were outlined in Section 2.9. Heterologous expression of isoprene monooxygenase. In this section, we describe the synthesis of isoA-F by PCR and subsequent restriction and ligation reactions to form pTipQC1:iso11. The original study describing the construction of pTipQC1 has now been cited on P10L417.